# Blow: a single-scale hyperconditioned flow for non-parallel raw-audio voice conversion

**Joan Serrà**
Telefónica Research
joan.serra@telefonica.com

**Santiago Pascual**
Universitat Politècnica de Catalunya
santi.pascual@upc.edu

**Carlos Segura**
Telefónica Research
carlos.seguraperales
@telefonica.com

## Abstract

End-to-end models for raw audio generation are a challenge, specially if they have to work with non-parallel data, which is a desirable setup in many situations. Voice conversion, in which a model has to impersonate a speaker in a recording, is one of those situations. In this paper, we propose Blow, a single-scale normalizing flow using hypernetwork conditioning to perform many-to-many voice conversion between raw audio. Blow is trained end-to-end, with non-parallel data, on a frame-by-frame basis using a single speaker identifier. We show that Blow compares favorably to existing flow-based architectures and other competitive baselines, obtaining equal or better performance in both objective and subjective evaluations. We further assess the impact of its main components with an ablation study, and quantify a number of properties such as the necessary amount of training data or the preference for source or target speakers.

## 1   Introduction

End-to-end generation of raw audio waveforms remains a challenge for current neural systems. Dealing with raw audio is more demanding than dealing with intermediate representations, as it requires a higher model capacity and a usually larger receptive field. In fact, producing high-level waveform structure was long thought to be intractable, even at a sampling rate of 16 kHz, and is only starting to be explored with the help of autoregressive models [1–3], generative adversarial networks [4, 5] and, more recently, normalizing flows [6, 7]. Nonetheless, generation without long-term context information still leads to sub-optimal results, as existing architectures struggle to capture such information, even if they employ a theoretically sufficiently large receptive field (cf. [8]).

Voice conversion is the task of replacing a source speaker identity by a targeted different one while preserving spoken content [9, 10]. It has multiple applications, the main ones being in the medical, entertainment, and education domains (see [9, 10] and references therein). Voice conversion systems are usually one-to-one or many-to-one, in the sense that they are only able to convert from a single or, at most, a handful of source speakers to a unique target one. While this may be sufficient for some cases, it limits their applicability and, at the same time, it prevents them from learning from multiple targets. In addition, voice conversion systems are usually trained with parallel data, in a strictly supervised fashion. To do so, one needs input/output pairs of recordings with the corresponding source/target speakers pronouncing the same underlying content with a relatively accurate temporal alignment. Collecting such data is non-scalable and, in the best of cases, problematic. Thus, researchers are shifting towards the use of non-parallel data [11–15]. However, non-parallel voice conversion is still an open issue, with results that are far from those using parallel data [10].

In this work, we explore the use of normalizing flows for non-parallel, many-to-many, raw-audio voice conversion. We propose Blow, a normalizing flow architecture that learns to convert voice recordings end-to-end with minimal supervision. It only employs individual audio frames, together with an

identifier or label that signals the speaker identity in such frames. Blow inherits some structure from Glow [16], but introduces several improvements that, besides yielding better likelihoods, prove crucial for effective voice conversion. Improvements include the use of a single-scale structure, many blocks with few flows in each, a forward-backward conversion mechanism, a conditioning module based on hypernetworks [17], shared speaker embeddings, and a number of data augmentation strategies for raw audio. We quantify the effectiveness of Blow both objectively and subjectively, obtaining comparable or even better performance than a number of baselines. We also perform an ablation study to quantify the relative importance of every new component, and assess further aspects such as the preference for source/target speakers or the relation between objective scores and the amount of training audio. We use public data and make our code available at `https://github.com/joansj/blow`. A number of voice conversion examples are provided in `https://blowconversions.github.io`.

## 2 Related work

To the best of our knowledge, there are no published works utilizing normalizing flows for voice conversion, and only three using normalizing flows for audio in general. Prenger et al. [6] and Kim et al. [7] concurrently propose using normalizing flows as a decoder from mel spectrograms to raw audio. Their models are based on Glow, but with a WaveNet [1] structure in the affine coupling network. Yamaguchi et al. [18] employ normalizing flows for audio anomaly detection and cross-domain image translation. They propose the use of class-dependant statistics to adaptively normalize flow activations, as done with AdaBN for regular networks [19].

### 2.1 Normalizing flows

Based on Barlow's principle of redundancy reduction [20], Redlich [21] and Deco and Brauer [22] already used invertible volume-preserving neural architectures. In more recent times, Dinh et al. [23] proposed performing factorial learning via maximum likelihood for image generation, still with volume-preserving transformations. Rezende and Mohamed [24] and Dinh et al. [25] introduced the usage of non-volume-preserving transformations, the formers adopting the terminology of normalizing flows and the use of affine and radial transformations [26]. Kingma and Dhariwal [16] proposed an effective architecture for image generation and manipulation that leverages $1 \times 1$ invertible convolutions. Despite having gained little attention compared to generative adversarial networks, autoregressive models, or variational autoencoders, flow-based models feature a number of merits that make them specially attractive [16], including exact inference and likelihood evaluation, efficient synthesis, a useful latent space, and some potential for gradient memory savings.

### 2.2 Non-parallel voice conversion

Non-parallel voice conversion has a long tradition of approaches using classical machine learning techniques [27–30]. However, today, neural networks dominate the field. Some approaches make use of automatic speech recognition or text representations to disentangle content from acoustics [31, 32]. This easily removes the characteristics of the source speaker, but further challenges the generator, which needs additional context to properly define the target voice. Many approaches employ a vocoder for obtaining an intermediate representation and as a generation module. Those typically convert between intermediate representations using variational autoencoders [11, 12], generative adversarial networks [13, 14], or both [15]. Finally, there are a few works employing a fully neural architecture on raw audio [33]. In that case, parts of the architecture may be pre-trained or not learned end-to-end. Besides voice conversion, there are some works dealing with non-parallel music or audio conversion: Engel et al. [34] propose a WaveNet autoencoder for note synthesis and instrument timbre transformations; Mor et al. [35] incorporate a domain-confusion loss for general musical translation and Nachmani and Wolf [36] incorporate an identity-agnostic loss for singing voice conversion; Haque et al. [37] use a sequence-to-sequence model for audio style transfer.

## 3 Flow-based generative models

Flow-based generative models learn a bijective mapping from input samples $\mathbf{x} \in \mathcal{X}$ to latent representations $\mathbf{z} \in \mathcal{Z}$ such that $\mathbf{z} = f(\mathbf{x})$ and $\mathbf{x} = f^{-1}(\mathbf{z})$. This mapping $f$, commonly called a normalizing flow [24], is a function parameterized by a neural network, and is composed by a

sequence of $k$ invertible transformations $f = f_1 \circ \cdots \circ f_k$. Thus, the relationship between $\mathbf{x}$ and $\mathbf{z}$, which are of the same dimensionality, can be expressed [16] as

$$\mathbf{x} \triangleq \mathbf{h}_0 \xleftrightarrow{f_1} \mathbf{h}_1 \xleftrightarrow{f_2} \mathbf{h}_2 \cdots \xleftrightarrow{f_k} \mathbf{h}_k \triangleq \mathbf{z}.$$

For a generative approach, we want to model the probability density $p(\mathcal{X})$ in order to be able to generate realistic samples. This is usually intractable in a direct way, but we can now use $f$ to model the exact log-likelihood

$$L(\mathcal{X}) = \frac{1}{|\mathcal{X}|} \sum_{i=1}^{|\mathcal{X}|} \log\left(p\left(\mathbf{x}_i\right)\right). \tag{1}$$

For a single sample $\mathbf{x}$, and using a change of variables, the inverse function theorem, compositionality, and logarithm properties (Appendix A), we can write

$$\log\left(p\left(\mathbf{x}\right)\right) = \log\left(p\left(\mathbf{z}\right)\right) + \sum_{i=1}^{k} \log\left|\det\left(\frac{\partial f_i(\mathbf{h}_{i-1})}{\partial \mathbf{h}_{i-1}}\right)\right|,$$

where $\partial f_i(\mathbf{h}_{i-1})/\partial \mathbf{h}_{i-1}$ is the Jacobian matrix of $f_i$ at $\mathbf{h}_{i-1}$ and the log-determinants measure the change in log-density made by $f_i$. In practice, one chooses transformations $f_i$ with triangular Jacobian matrices to achieve a fast calculation of the determinant and ensure invertibility, albeit these may not be as expressive as more elaborate ones (see for instance [38–40]). Similarly, one chooses an isotropic unit Gaussian for $p(\mathbf{z})$ in order to allow fast sampling and straightforward operations.

A number of structures and parameterizations of $f$ and $f_i$ have been proposed for image generation, the most popular ones being RealNVP [25] and Glow [16]. More recently, other works have proposed improvements for better density estimation and image generation in multiple contexts [38–43]. RealNVP uses a block structure with batch normalization, masked convolutions, and affine coupling layers. It combines those with $2{\times}2$ squeezing operations and alternating checkerboard and channel-wise masks. Glow goes one step further and, besides replacing batch normalization by activation normalization (ActNorm), introduces a channel-wise mixing through invertible $1{\times}1$ convolutions. Its architecture is composed of 3 to 6 blocks, formed by a $2{\times}2$ squeezing operation and 32 to 64 steps of flow, which comprise a sequence of ActNorm, $1{\times}1$ invertible convolution, and affine coupling. For the affine coupling, three convolutional layers with rectified linear units (ReLUs) are used. Both Glow and RealNVP feature a multi-scale structure that factors out components of $\mathbf{z}$ at different resolutions, with the intention of defining intermediary levels of representation at different granularities. This is also the strategy followed by other image generation flows and the two existing audio generation ones [6, 7].

## 4   Blow

Blow inherits some structure from Glow, but incorporates several modifications that we show are key for effective voice conversion. The main ones are the use of (1) a single-scale structure, (2) more blocks with less flows in each, (3) a forward-backward conversion mechanism, (4) a hyperconditioning module, (5) shared speaker embeddings, and (6) a number of data augmentation strategies for raw audio. We now provide an overview of the general structure (Fig. 1).

We use one-dimensional $2\times$ squeeze operations with an alternate pattern [25] and a series of steps of flow (Fig. 1, left). A step of flow is composed of a linear invertible layer as channel mixer (similar to a $1{\times}1$ invertible convolution in the two-dimensional case), ActNorm, and a coupling network with affine coupling (Fig. 1, center). Coupling networks are formed by one-dimensional convolutions and hyperconvolutions with ReLU activations (Fig. 1, right). The last convolution and the hyperconvolution of the coupling network have a kernel width of 3, while the intermediate convolution has a kernel width of 1 (we use $512{\times}512$ channels). The same speaker embedding feeds all coupling networks, and is independently adapted for each hyperconvolution. Following common practice, we compare the output $\mathbf{z}$ against a unit isotropic Gaussian and optimize the log-likelihood $L$ (Eq. 1) normalized by the dimensionality of $\mathbf{z}$.

### 4.1   Single-scale structure

Besides the aforementioned ability to deal with intermediary levels of representation, a multi-scale structure is thought to encourage the gradient flow and, therefore, facilitate the training of very deep

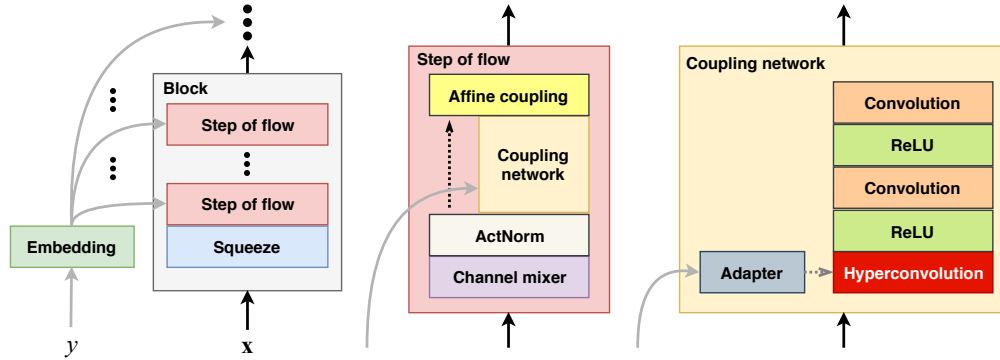

Figure 1: Blow schema featuring its block structure (left), steps of flow (center), and coupling network with hyperconvolution module (right).

models [44] like normalizing flows. Here, in preliminary analysis, we observed that speaker identity traits were almost present only at the coarser level of representation. Moreover, we found that, by removing the multi-scale structure and carrying the same input dimensionality across blocks, not only gradients were flowing without issue, but better log-likelihoods were also obtained (see below).

We believe that the fact that gradients still flow without factoring out block activations is because the log-determinant term in the loss function is still factored out at every flow step (Appendix A). Therefore, some gradient is still shuttled back to the corresponding layer and below. The fact that we obtain better log-likelihoods with a single-scale structure was somehow expected, as block activations now undergo further processing in subsequent blocks. However, to our understanding, this aspect seems to be missed in the likelihood-based evaluation of current image generation flows.

## 4.2   Many blocks

Flow-based image generation models deal with images between $32\times32$ and $256\times256$ pixels. For raw audio, a one-dimensional input of 256 samples at 16 kHz corresponds to 16 ms, which is insufficient to capture any interesting speech construct. Phoneme duration can be between 50 and 180 ms [45], and we need a little more length to model some phoneme transition. Therefore, we need to increase the input and the receptive field of the model. To do so, flow-based audio generation models [6, 7] opt for more aggressive squeezing factors, together with a WaveNet-style coupling network with dilation up to $2^8$. In Blow, in contrast, we opt for using many blocks with relatively few flow steps each. In particular, we use 8 blocks with 12 flows each (an $8\times12$ structure). Since every block has a $2\times$ squeeze operation, this implies a total squeezing of $2^8$ samples.

Considering two convolutions of kernel width 3, an $8\times12$ structure yields a receptive field of roughly 12500 samples that, at 16 kHz, corresponds to 781 ms. However, to allow for larger batch sizes, we use an input frame size of 4096 samples (256 ms at 16 kHz). This is sufficient to accommodate, at least, one phoneme and one phoneme transition if we cut in the middle of words, and is comparable to the receptive field of other successful models like WaveNet. Blow operates on a frame-by-frame basis without context; we admit that this could be insufficient to model long-range speaker-dependent prosody, but nonetheless believe it is enough to model core speaker identity traits.

## 4.3   Forward-backward conversion

The default strategy to perform image manipulation [16] or class-conditioning [41, 42] in Glow-based models is to work in the $\mathbf{z}$ space. This has a number of interesting properties, including the possibility to perform progressive changes or interpolations, and the potential for few-shot learning or manipulations based on small data. However, we observed that, for voice conversion, results following this strategy were largely unsatisfactory (Appendix B).

Instead of using $\mathbf{z}$ to perform identity manipulations, we think of it as an identity-agnostic representation. Our idea is that any supplied condition specifying some real input characteristic of $\mathbf{x}$ should be useful to transform $\mathbf{x}$ to $\mathbf{z}$, specially if we consider a maximum likelihood objective. That is, knowing

a condition/characteristic of the input should facilitate the discovery of further similarities that were hidden by said condition/characteristic, and thus facilitate learning. Following this line of thought, if conditioning at multiple levels in the flow from $\mathbf{x}$ to $\mathbf{z}$ progressively get us to a condition-free $\mathbf{z}$ space (Appendix C.3), then, when transforming back from $\mathbf{z}$ to $\mathbf{x}$ with a different condition, that should also progressively imprint the characteristics of this new condition to the output $\mathbf{x}$. Blow uses the source speaker identifier $y_{\mathrm{S}}$ for transforming $\mathbf{x}^{(\mathrm{S})}$ to $\mathbf{z}$, and the target speaker identifier $y_{\mathrm{T}}$ for transforming $\mathbf{z}$ to the converted audio frame $\mathbf{x}^{(\mathrm{T})}$.

## 4.4 Hyperconditioning

A straightforward place to introduce conditioning in flow-based models is the coupling network, as no Jacobian matrix needs to be computed and no invertibility constraints apply. Furthermore, in the case of affine channel-wise couplings [16, 25], the coupling network is in charge of performing most of the transformation, so we want it to have a great representation power, possibly boosted by further conditioning information. A common way to condition the coupling network is to add or concatenate some representation to its input layers. However, based on our observations that concatenation tended to be ignored and that addition was not powerful enough, we decided to perform conditioning directly with the weights of the convolutional kernels. That is, that a conditioning representation determines the weights employed by a convolution operator, like done with hypernetworks [17]. We do it at the first layer of the coupling network (Fig. 1, right).

Using one-dimensional convolutions, and given an input activation matrix $\mathbf{H}$, for the $i$-th convolutional filter we have

$$\mathbf{h}^{(i)} = \mathbf{W}_y^{(i)} * \mathbf{H} + b_y^{(i)}, \tag{2}$$

where $*$ is the one-dimensional convolution operator, and $\mathbf{W}_y^{(i)}$ and $b_y^{(i)}$ represent the $i$-th kernel weights and bias, respectively, imposed by condition $y$. A set of $n$ condition-dependent kernels and biases $\mathcal{K}_y$ can be obtained by

$$\mathcal{K}_y = \left\{ \left( \mathbf{W}_y^{(1)}, b_y^{(1)} \right) \dots \left( \mathbf{W}_y^{(n)}, b_y^{(n)} \right) \right\} = g\left( \mathbf{e}_y \right), \tag{3}$$

where $g$ is an adapter network that takes the conditioning representation $\mathbf{e}_y$ as input, which in turn depends on condition identifier $y$ (the speaker identity in our case). Vector $\mathbf{e}_y$ is an embedding that can either be fixed or initialized at some pre-calculated feature representation of a speaker, or learned from scratch if we need a standalone model. In this paper we choose the standalone version.

## 4.5 Structure-wise shared embeddings

We find that learning one $\mathbf{e}_y$ per coupling network usually results in sub-optimal results. We hypothesize that, given a large number of steps of flow (or coupling networks), independent conditioning representations do not need to focus on the essence of the condition (the speaker identity), and are thus free to learn any combination of numbers that minimizes the negative log-likelihood, irrespective of their relation with the condition. Therefore, to reduce the freedom of the model, we decide to constrain such representations. Loosely inspired by the StyleGAN architecture [46], we set a single learnable embedding $\mathbf{e}_y$ that is shared by each coupling network in all steps of flow (Fig. 1, left). This reduces both the number of parameters and the freedom of the model, and turns out to yield better results. Following a similar reasoning, we also use the smallest possible adapter network $g$ (Fig. 1, right): a single linear layer with bias that merely performs dimensonality adjustment.

## 4.6 Data augmentation

To train Blow, we discard silent frames (Appendix B) and then enhance the remaining ones with 4 data augmentation strategies. Firstly, we apply a temporal jitter. We shift the start $j$ of each frame $\mathbf{x}$ as $j' = j + \lfloor U(-\xi, \xi) \rfloor$, where $U$ is a uniform random number generator and $\xi$ is half of the frame size. Secondly, we use a random pre-/de-emphasis filter. Since the identity of the speaker is not going to vary with a simple filtering strategy, we apply an emphasis filter [47] with a coefficient $\alpha = U(-0.25, 0.25)$. Thirdly, we perform a random amplitude scaling. Speaker identity is also going to be preserved with scaling, plus we want the model to be able to deal with any amplitude between $-1$ and $1$. We use $\mathbf{x}' = U(0, 1) \cdot \mathbf{x} / \max(|\mathbf{x}|)$. Finally, we randomly flip the values in the frame. Auditory perception is relative to an average pressure level, so we can flip the sign of $\mathbf{x}$ to obtain a different input with the same perceptual qualities: $\mathbf{x}' = \mathrm{sgn}(U(-1, 1)) \cdot \mathbf{x}$.

### 4.7 Implementation details

We now outline the details that differ from the common implementation of flow-based generative models and further refer the interested reader to the provided code for a full account of them. We also want to note that we did not perform any hyperparameter tuning on Blow.

**General —** We train Blow with Adam using a learning rate of $10^{-4}$ and a batch size of 114. We anneal the learning rate by a factor of 5 if 10 epochs have passed without improvement in the validation set, and stop training at the third time this happens. We use an $8 \times 12$ structure, with $2 \times$ alternate-pattern squeezing operations. For the coupling network, we split channels into two halves, and use one-dimensional convolutions with 512 filters and kernel widths 3, 1, and 3. Embeddings are of dimension 128. We train with a frame size of 4096 at 16 kHz with no overlap, and initialize the ActNorm weights with one data-augmented batch (batches contain a random mixture of frames from all speakers). We synthesize with a Hann window and 50% overlap, normalizing the entire utterance between $-1$ and 1. We implement Blow using PyTorch [48].

**Coupling —** As done in the official Glow code (but not mentioned in the paper), we find that constraining the scaling factor that comes out of the coupling network improves the stability of training. For affine couplings with channel-wise concatenation

$$\mathbf{H}' = \left[ \; \mathbf{H}_{1:c} \; , \; s'(\mathbf{H}_{1:c}) \left( \mathbf{H}_{c+1:2c} + t(\mathbf{H}_{1:c}) \right) \; \right],$$

where $2c$ is the total number of channels, we use

$$s'(\mathbf{H}_{1:c}) = \sigma(s(\mathbf{H}_{1:c}) + 2) + \epsilon,$$

where $\sigma$ corresponds to the sigmoid function and $\epsilon$ is a small constant to prevent an infinite log-determinant (and division by 0 in the reverse pass).

**Hyperconditioning —** If we strictly follow Eqs. 2 and 3, the hyperconditioning operation can involve both a large GPU memory footprint ($n$ different kernels per batch element) and time-consuming calculations (a double loop for every kernel and batch element). This can, in practice, make the operation impossible to perform for a very deep flow-based architecture like Blow. However, by restricting the dimensionality of kernels $\mathbf{W}_y^{(i)}$ such that every channel is convolved with its own set of kernels, we can achieve a minor GPU footprint and a tractable number of parameters per adaptation network. This corresponds to depthwise separable convolutions [49], and can be implemented with grouped convolution [50], available in most deep learning libraries.

## 5 Experimental setup

To study the performance of Blow we use the VCTK data set [51], which comprises 46 h of audio from 109 speakers. We downsample it at 16 kHz and randomly extract 10% of the sentences for validation and 10% for testing (we use a simple parsing script to ensure that the same sentence text does not get into different splits, see Appendix B). With this amount of data, the training of Blow takes 13 days using three GeForce RTX 2080-Ti GPUs[1]. Conversions are performed between all possible gender combinations, from test utterances to randomly-selected VCTK speakers.

To compare with existing approaches, we consider two flow-based generative models and two competitive voice conversion systems. As flow-based generative models we adapt Glow [16] to the one-dimensional case and replicate a version of Glow with a WaveNet coupling network following [6, 7] (Glow-WaveNet). Conversion is done both via manipulation of the $\mathbf{z}$ space and by learning an identity conditioner (Appendix B). These models use the same frame size and have the same number of flow steps as Blow, with a comparable number of parameters. As voice conversion systems we implement a VQ-VAE architecture with a WaveNet decoder [33] and an adaptation of the StarGAN architecture to voice conversion like StarGAN-VC [14]. VQ-VAE converts in the waveform domain, while StarGAN does it between mel-cepstrums. Both systems can be considered as very competitive for the non-parallel voice conversion task. We do not use pre-training nor transfer learning in any of the models.

To quantify performance, we carry out both objective and subjective evaluations. As objective metrics we consider the per-dimensionality log-likelihood of the flow-based models ($L$) and a spoofing

Table 1: Objective scores and their relative difference for possible Blow alternatives (5 min per speaker, 100 epochs).

| Configuration | $L$ [nat/dim] | Spoofing [%] |
|---|---|---|
| Blow | **4.30** | **66.2** |
| 1: with 3×32 structure | 4.01 (− 6.7%) | 17.2 (−74.0%) |
| 2: with 3×32 structure (squeeze of 8) | 4.21 (− 2.1%) | 65.7 (− 0.8%) |
| 3: with multi-scale structure | 3.64 (−15.3%) | 3.5 (−94.7%) |
| 4: with multi-scale structure (5×19, squeeze of 4) | 3.99 (− 7.2%) | 16.6 (−74.9%) |
| 5: with additive conditioning (coupling network) | 4.28 (− 0.5%) | 39.5 (−40.3%) |
| 6: with additive conditioning (before ActNorm) | 4.28 (− 0.5%) | 22.5 (−66.0%) |
| 7: without data augmentation | 4.15 (− 3.5%) | 28.3 (−57.2%) |

Table 2: Objective and subjective voice conversion scores. For all measures, higher is better. The first two reference rows correspond to using original recordings from source or target speakers as target.

| Approach | Objective | | Subjective | |
|---|---|---|---|---|
| | $L$ [nat/dim] | Spoofing [%] | Naturalness [1–5] | Similarity [%] |
| Source as target | n/a | 1.1 | 4.83 | 10.6 |
| Target as target | n/a | 99.3 | 4.83 | 98.5 |
| Glow | 4.11 | 1.2 | n/a | n/a |
| Glow-WaveNet | 4.18 | 3.1 | n/a | n/a |
| StarGAN | n/a | 44.4 | **2.87** | 61.8 |
| VQ-VAE | n/a | 65.0 | 2.42 | 69.7 |
| Blow | **4.45** | **89.3** | 2.83 | **77.6** |

measure reflecting the percentage of times a conversion is able to fool a speaker identification classifier (Spoofing). The classifier is an MFCC-based single-layer classifier trained with the same split as the conversion systems (Appendix B). For the subjective evaluation we follow Wester et al. [52] and consider the naturalness of the speech (Naturalness) and the similarity of the converted speech to the target identity (Similarity). Naturalness is based on a mean opinion score from 1 to 5, while Similarity is an aggregate percentage from a binary rating. A total of 33 people participated in the subjective evaluation. Further details on our experimental setup are given in Appendix B.

## 6 Results

### 6.1 Ablation study

First of all, we assess the effect of the introduced changes with objective scores $L$ and Spoofing. Due to computational constraints, in this set of experiments we limit training to 5 min of audio per speaker and 100 epochs. The results are in Table 1. In general, we see that all introduced improvements are important, as removing any of them always implies worse scores. Nonetheless, some are more critical than others. The most critical one is the use of a single-scale structure. The two alternatives with a multi-scale structure (3–4) yield the worst likelihoods and spoofings, to the point that (3) does not even perform any conversion. Using an 8×12 structure instead of the original 3×32 structure of Glow can also have a large effect (1). However, if we further tune the squeezing factor we can mitigate it (2). Substituting the hyperconditioning module by a regular convolution plus a learnable additive embedding has a marginal effect on $L$, but a crucial effect on Spoofing (5–6). Finally, the proposed data augmentation strategies also prove to be important, at least with 5 min per speaker (7).

### 6.2 Voice conversion

In Table 2 we show the results for both objective and subjective scores. The two objective scores, $L$ and Spoofing, indicate that Blow outperforms the other considered approaches. It achieves a relative $L$ increment of 6% from Glow-Wavenet and a relative Spoofing increment of 37% from VQ-VAE. Another thing to note is that adapted Glow-based models, although achieving a reasonable likelihood,

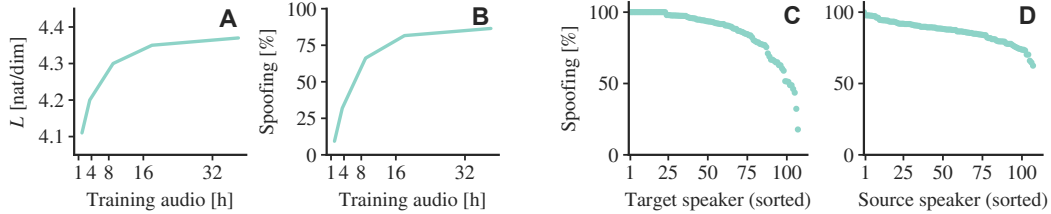

Figure 2: Objective scores with respect to amount of training (A–B) and target/source speaker (C–D).

are not able to perform conversion, as their Spoofing is very close to that of the "source as target" reference. Because of that, we discarded those in the subjective evaluation.

The subjective evaluation confirms the good performance of Blow. In terms of Naturalness, StarGAN outperforms Blow, albeit by only a 1% relative difference, without statistical significance (ANOVA, $p = 0.76$). However, both approaches are significantly below the reference audios ($p < 0.05$). In terms of similarity to the target, Blow outperforms both StarGAN and VQ-VAE by a relative 25 and 11%, respectively. Statistical significance is observed between Blow and StarGAN (Barnard's test, $p = 0.02$) but not between Blow and VQ-VAE ($p = 0.13$). Further analysis of the obtained subjective scores can be found in Appendix C. To put Blow's results into further perspective, we can have a look at the non-parallel task of the last voice conversion challenge [10], where systems that do not perform transfer learning or pre-training achieve Naturalness scores slightly below 3.0 and Similarity scores equal to or lower than 75%. Example conversions can be listened from https://blowconversions.github.io.

### 6.3    Amount of training data and source/target preference

To conclude, we study the behavior of the objective scores when decreasing the amount of training audio (including the inherent silence in the data set, which we estimate is around 40%). We observe that, at 100 epochs, training with 18 h yields almost the same likelihood (Fig. 2A) and spoofing (Fig. 2B) than training with the full set of 37 h. With it, we do not observe any clear relationship between Spoofing and per-speaker duration (Appendix C). What we observe, however, is a tendency with regard to source and target identities. If we average spoofing scores for a given target identity, we obtain both almost-perfect scores close to 100% and some scores below 50% (Fig. 2C). In contrast, if we average spoofing scores for a given source identity, those are almost always above 70% and below 100% (Fig. 2D). This indicates that the target identity is critical for the conversion to succeed, with relative independence of the source. We hypothesize that this is due to the way normalizing flows are trained (maximizing likelihood only for single inputs and identifiers; never performing an actual conversion to a target speaker), but leave the analysis of this phenomenon for future work.

## 7    Conclusion

In this work we put forward the potential of flow-based generative models for raw audio synthesis, and specially for the challenging task of non-parallel voice conversion. We propose Blow, a single-scale hyperconditioned flow that features a many-block structure with shared embeddings and performs conversion in a forward-backward manner. Because Blow departs from existing flow-based generative models in these aspects, it is able to outperform those and compete with, or even improve upon, existing non-parallel voice conversion systems. We also quantify the impact of the proposed improvements and assess the effect that the amount of training data and the selection of source/target speaker can have in the final result. As future work, we want to improve the model to see if we can deal with other tasks such as speech enhancement or instrument conversion, perhaps by further enhancing the hyperconditioning mechanism or, simply, by tuning its structure or hyperparameters.

**Acknowledgments**

We are grateful to all participants of the subjective evaluation for their input and feedback. We thank Antonio Bonafonte, Ferran Diego, and Martin Pielot for helpful comments. SP acknowledges partial support from the project TEC2015-69266-P (MINECO/FEDER, UE).

## Footnotes

[1]Nonetheless, conversion plus synthesis with 1 GPU and 50% overlap is around $14 \times$ faster than real time.

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
