[Supplementary Material]

# Blow: a single-scale hyperconditioned flow for non-parallel raw-audio voice conversion

**Joan Serrà**
Telefónica Research
joan.serra@telefonica.com

**Santiago Pascual**
Universitat Politècnica de Catalunya
santi.pascual@upc.edu

**Carlos Segura**
Telefónica Research
carlos.seguraperales
@telefonica.com

# Appendix

## A    Recap of the log-likelihood equation derivation

Following Rezende and Mohamed [1], if we use a normalizing flow $f$ to transform a random variable $\mathbf{x}$ with distribution $p(\mathbf{x})$, the resulting random variable $\mathbf{z} = f(\mathbf{x})$ has a distribution

$$p\left(\mathbf{z}\right) = p\left(f\left(\mathbf{x}\right)\right) = p\left(\mathbf{x}\right) \left| \det \left( \frac{\partial f^{-1}(\mathbf{z})}{\partial \mathbf{z}} \right) \right|,$$

which is derived from the change of variables formula. By the inverse function theorem, we can work with the Jacobian of $f$,

$$p\left(\mathbf{z}\right) = p\left(\mathbf{x}\right) \left| \det \left( \frac{\partial f(\mathbf{z})}{\partial \mathbf{z}} \right) \right|^{-1}$$

and, taking logarithms and rearranging, we reach

$$\log\left(p\left(\mathbf{x}\right)\right) = \log\left(p\left(\mathbf{z}\right)\right) + \log \left| \det \left( \frac{\partial f(\mathbf{z})}{\partial \mathbf{z}} \right) \right|,$$

as expressed by, for instance, Dinh et al. [2]. Finally, since $f$ is a composite function (Sec. 3), we can write the previous equation as Kingma and Dhariwal [3]:

$$\log\left(p\left(\mathbf{x}\right)\right) = \log\left(p\left(\mathbf{z}\right)\right) + \sum_{i=1}^{k} \log \left| \det \left( \frac{\partial f_i(\mathbf{h}_{i-1})}{\partial \mathbf{h}_{i-1}} \right) \right|.$$

This is the expression we use to optimize the normalizing flow. Notice that log-determinants can be factored out at each flow step, shuttling gradients back to each $f_i$ (or $\mathbf{h}_i$) and below.

## B    Detail of the experimental setup

### B.1    Data

As mentioned in the main paper, we use the VCTK data set [4], which originally contains 46 h of audio from 109 speakers. The only pre-processing we perform to the original audio is to downsample to 16 kHz and to normalize every file between $-1$ and 1. Later, at training time, silent frames with a standard deviation below 0.025 are discarded. As mentioned, we use frames of 4096 samples.

To obtain train, validation, and test splits, we parse the text of every sentence and group utterances of the same text (we discard the speaker without available text). We then randomly extract 10% of the

sentences for validation and 10% for test. This way, we force that the same text content is not present in more than one split, and therefore that sentences in validation or test are not included in training. The total amount of training audio is 36 h which, with 108 speakers, yields an average of 20 min per speaker. Other statistics are reported in Table 1.

Table 1: Train, validation, and test numbers.

| Description | Train | Validation | Test |
|---|---|---|---|
| Audio duration | 36.7 h | 4.4 h | 4.5 h |
| Number of sentences | 10609 | 1325 | 1325 |
| Number of files | 35247 | 4417 | 4406 |
| Number of frames (discarding silence) | 291154 | 34390 | 35253 |

All reported results are based on the test split, including the audios used for subjective evaluation. We perform one conversion per test file, by choosing a different speaker from the pool of all available speakers uniformly at random (irrespective of the gender or other metadata).

## B.2 Baselines

### B.2.1 Glow-based

For performing audio conversion with Glow-based baselines, we initially considered a conditioning-only strategy. In the case of Glow, this implied computing a Gaussian mean for every label at training time and subtracting it in $\mathbf{z}$ space (adding it when going from $\mathbf{z}$ to $\mathbf{x}$). In the case of Glow-WaveNet, as it directly accepts a conditioning, we implement independent learnable embeddings that are added at the first layer of every coupling, as done with the mel conditioning of WaveGlow [5]. The conditioning-only strategy, however, turned out to perform poorly for these models in preliminary experiments.

Using a manipulation-only strategy as proposed by Kingma and Dhariwal [3] was also found to perform poorly. Some conversion could be perceived, like for instance changing identities from male to female, but obtained identities were not similar to the target ones. In addition, we found annoying audio artifacts were easily appearing, and that those could be amplified with just minimal changes in the manipulation strategy.

In the end, we decided to use both strategies and augment the conditioning-only strategy with the semantic manipulation one. We empirically chose a scaling factor of 3 as a trade-off between amount of conversion and generation of artifacts. We also found that weighting the contribution to the mean by the energy of $\mathbf{x}$ could slightly improve conversion.

### B.2.2 StarGAN

The baseline StarGAN model is based on StarGAN-VC [6], which uses StarGAN [7] to learn non-parallel many-to-many mappings between speakers. It is worth noting that this approach does not work at the waveform level, but instead extracts the fundamental frequency, aperiodicity, and spectral envelope from each audio clip, and then performs the conversion by means of its generator at the spectral envelope level. For generating the target speakers' speech, the WORLD vocoder [8] is used with the transformed spectral envelope, linearly-converted pitch, and original aperiodicity as inputs.

In the original StarGAN-VC paper, the experiments comprised only 4 speakers (2 male and 2 female), while in this work we extended it to all VCTK speakers. However, in our setup, publicly available implementations of this architecture did not generate a reasonably natural speech, and hence we tried with a alternative implementation. In particular, our baseline is based on an implementation[1] that uses the same architecture as the original image-to-image StarGAN [7]. The main difference with StarGAN-VC is that it does not include any conditioning on the speaker in the discriminator network, but instead the discriminator and domain classifier (that acts as a speaker classifier) share the same underlying network weights. The other difference is that the training uses a Wasserstein GAN objective with gradient penalty.

### B.2.3 VQ-VAE

The baseline VQ-VAE model for voice conversion is based on [9]. The exact specification details such as the waveform encoder architecture are not provided in the paper and, to our knowledge, an official model implementation has not been published so far. We tried a number of non-official implementations but, in the end, found our own implementation to perform better in preliminary experiments.

Our implementation follows as closely as possible [9]. We use 7 strided convolutions for the audio encoder with a stride of 2 and a kernel size of 4, with 448 channels in the last layer. Therefore, we have a time-domain compression of $2^7$ compared to the original raw audio. The feature map is then projected into a latent space of dimension 128, and the discrete space of the quantized vectors is 512. The discrete latent codes are concatenated with the target speaker embedding and then upsampled in the time dimension using a deconvolutional layer with a stride of $2^7$, which is used as the local conditioning for the WaveNet decoder. To speed up the audio generation, our WaveNet implementation uses the one provided by NVIDIA[2], which implements the WaveNet variant described by Arik et al. [10]. However, to perform a fair comparison with Blow (and possibly differently from [9]), we do not use any pre-trained weight in the WaveNet nor the VQ-VAE structures.

### B.3 Spoofing classifier

To objectively evaluate the capacity of the considered approaches to perform voice conversion we employ a speaker identity classifier, which we train on the same split as the conversion approaches. The classifier uses classic speech features computed within a short-time frame. With that, we believe the Spoofing measure captures not only speaker identities, but can also be affected by audio artifacts or distortions that may have an impact to the short-time, frame-based features. We use 40 mel-frequency cepstral coefficients (MFCCs), their deltas, their delta-deltas, and the root mean square energy. From those we then compute the mean and the standard deviation across frames to summarize the speaker identity in an audio file. To extract features we use librosa[3] with default parameters, except for some of the MFCC ones: FFT hop of 128, FFT window of 256, FFT size of 2048, and 200 mel bands. After feature extraction, we apply z-score normalization, computing the mean and the standard deviation from training data.

The classifier is a linear network with dropout, trained with categorical cross-entropy. We then apply a dropout of 0.4 to the input features and a linear layer with bias. We train the classifier with Adam using a learning rate of $10^{-3}$ and stop training when the validation loss has not improved for 10 epochs. With 108 speakers, this classifier achieves an accuracy of 99.3% on the test split.

### B.4 Subjective evaluation

For the subjective evaluation we follow Wester et al. [11]. They divide it into two aspects: naturalness and similarity. Naturalness aims to measure the amount of artifacts or distortion that is present in the generated signals. Similarity aims to measure how much the converted speaker identity resembles either the source or the target identity. Naturalness is measured with a mean opinion score between 1 and 5, and similarities are measured with a binary decision, allowing the option to express some uncertainty. Statistical significance is assessed with an analysis of variance (ANOVA) for Naturalness and with Barnard's test for Similarity (both single tail, with $p < 0.05$).

A total of 33 subjects participated of the subjective evaluation. From those, 3 were native English speakers and 8 declared having some speech processing expertise. Participants were presented to 16 audio examples in the Naturalness assessment part (4 per system) and to 16 audio pairs in the Similarity assessment part (4 per system, two for similar to the target and two for similar to the source assessments).

# C   Additional results

## C.1   Analysis of subjective scores

A visual summary of the numbers reported in the main paper is depicted in Fig. 1. We see that the three considered systems cluster together, falling apart from the real target and source voices. Among the three systems, Blow stands out, specially in similarity to the target (vertical axis), and competes closely with StarGAN in terms of Naturalness (horizontal axis).

Figure 1: Scatter plot of the subjective evaluation results: Naturalness (horizontal axis) and similarity to the target (vertical axis) for the considered models and references.

Figure 2: Box plot of Naturalness MOS. Red triangles indicate the arithmetic mean.

Figure 3: Similarity to the target ratings disregarding confidence (left) and including confidence assessment (right).

Figure 4: Similarity to the source ratings disregarding confidence (left) and including confidence (right) assessments.

Table 2: Objective scores at 100 epochs for different training sizes.

| Total amount of training audio | 1.8 h | 3.6 h | 9 h | 18 h | 37 h (full) |
|---|---|---|---|---|---|
| Training audio per speaker | 1 min | 2 min | 5 min | 10 min | 20.4 min (average) |
| $L$ [nat/dim] | 4.11 | 4.20 | 4.30 | 4.35 | 4.37 |
| Spoofing [%] | 9.3 | 31.9 | 66.2 | 81.6 | 86.5 |

If we study naturalness scores alone, we see that the difference between Blow and StarGAN is minimal (Fig 2). Actually, we find no statistically significant difference between the two (ANOVA, $p = 0.76$). This is a good result if we consider that spectral-based approaches, such as StarGAN, are often preferred with regard to Naturalness due to their constriction to not generate audible artifacts in the time domain.

If we study similarity judgments alone, we observe a different picture (Figs. 3 and 4). Focusing on similarity to the target, Blow performs better than StarGAN and VQ-VAE. The ranking of the methods can be clearly seen when disregarding the confidence on the decisions (Fig. 3, left). Statistical significance is observed between Blow and StarGAN (Barnard's test, $p = 0.02$) but not between Blow and VQ-VAE ($p = 0.13$). If we consider the degree of confidence, we see the difference is in the "Same: not sure" ratings, as all three obtain almost the same number of "Same: absolutely sure" decisions (Fig. 3, right).

Finally, it is also interesting to look at the results for similarity to the source (Fig. 4). In them, we see that Blow and StarGAN generate slightly more audios that are considered to be similar to the source than VQ-VAE. This could indicate a problem in converting from some source identities, as the characteristics of those seem to remain in the conversion. However, in general, the amount of "Similar to the source" conversions is low, below 20%, and relatively close to the 10.6% obtained for the control group that compares real different identities (the target ones) with the source identities (Fig. 4, leftmost bars).

## C.2    Amount of training audio

For completeness, in Table 2 we report the exact numbers depicted in Figs. 2A and 2B of the main paper. In Fig. 5, we further study Spoofing with respect to the amount of audio per speaker in the full training set. We do not observe any trend with respect to duration of training audio per speaker. All these results are calculated after 100 epochs of training.

## C.3    Condition-free latent space

A driving idea of Blow is that the latent space $\mathbf{z}$ should be condition-free (or identity-agnostic). This is what motivates us to use hyperconditioning to progressively remove condition/identity characteristics when transforming from $\mathbf{x}$ to $\mathbf{z}$ (and later to progressively imprint new condition/identity characteristics when transforming back from $\mathbf{z}$ to $\mathbf{x}$). In order to substantiate a bit more our hypothesis, we decide to study the capacity to perform speaker identification in the latent space $\mathbf{z}$. The idea is that,

Figure 5: Spoofing percentage with respect to amount of training audio per speaker at 100 training epochs (full data set, including silence).

if **z** vectors contain some speaker information, a classifier should be able to perform some speaker identification in **z** space.

To quantify the amount of speaker identity information present in **z**, we proceed as with the Spoofing classifier (see above), but using the actual vectors **z** as frame-based features. The only difference is that, in the current case, we are interested in the result of a more complex classifier with enough power to extract non-trivial, relevant information from the features, if any. To this end, we consider a random forest classifier and a multi-layer perceptron (we use scikit-learn version 0.20.2 with default parameters, except for the number of estimators of the random forest classifier, which we set to 50, and the number of layers of the multi-layer perceptron, which we set to three, with 1000 and 100 intermediate activations).

The test accuracies we obtain for the two classifiers are 1.8% (random forest) and 1.4% (multi-layer perceptron). Both are only marginally above random chance (1.1%), and far from the value obtained by classic features extracted from **x** (99.3%). This gives us an indication that there is little identity information in the **z** space. However, to further confirm our original hypothesis, additional experiments on the latent space **z**, which are beyond the scope of the current work, should be carried out.

## Footnotes

[1]https://github.com/liusongxiang/StarGAN-Voice-Conversion

[2]https://github.com/NVIDIA/nv-wavenet

[3]http://librosa.github.io/librosa