[Reviews · NeurIPS 2019]

Reviewer 1



The paper is very well written in clear English. The proposed framework is methodologically sound and sufficient details are given to make it fully reproducible and extendable. The experiments are convincing, with a relevant choice of comparative methods and an insightful ablation study. The bibliography and litterature review are adequate. Minor remarks and typos: -line 96: "albeit these may be-not be as expressive" -> may not be as expressive -Sec; 4.2.: it is not obvious how the proposed architecture yields a receptive field of 12500, since the total squeezing is 2^8. Some more details on this would be appreciated, as well as on the final dimension of z (which should be 4096, but this is implicit and hard to deduce). -In Sec. 4.5, the authors refer to "network g", but this does not appear in Figure 1. Although one may deduce that this corresponds to the "embedding" green block, this could be made more clear. -In Sec. 4.7, more details would be appreciated on how btaches are built . In particular, do they contain a mix of frames from all speakers, from a subset of speakers or from a single speaker? -The authors do not provide error bars for any of their results. Some standard deviations would be appreciated.

Reviewer 2



Originality: The authors propose original work, as they are the first to utilize normalizing flows for voice conversion. Quality: The paper provided extensive experiments as well as ablation studies, and provided samples of the voice conversions. Despite the lack of baselines, the authors did a good job deconstructing which components of their model were important and also experimenting with other, non-trivial models such as StarGAN and VQ-VAE. One thing I noticed, though, was when the source and target voices were of different genders, the converted voice seemed to have more artifacts and noise than usual. Clarity: The paper was well-written and easy to follow, which I appreciated. Significance: There is not a significant amount of work in non-parallel music or audio conversion in general, so this work presents a step forward in opening up new research directions. ----------------------------------------------------------------- UPDATE: I appreciate the authors addressing my confusion regarding the forward-backward conversion section, as well as the additional experiment about the identity neutrality of the z space. I will keep my original score.

Reviewer 3



Pros: 1. Using the invertibility(forward-backward) of flow-based model to do voice conversion is an overall clever idea. 2. End-to-end training of voice conversion system directly on raw waveform is elegant. Cons: 1. The novelty in machine learning/deep learning is limited. The session 4 is more like an architecture tuning summary of Glow/WaveGlow. Also the quality of posted audio samples and subjective evaluation(both naturalness and similarity) need to be improved. 2. For flow-based model, the mapping between latent z and data x has to be bijective. So there is no information bottleneck like auto-encoder based models(e.g. VQ-VAE). Thus, it's hard to convince others that z is in condition-neutral space like the author proposed in session 4.3, especially when the speaker embeddings are not powerful enough. Further experiments, like latent space analysis in VQ-VAE, will be necessary to prove z is condition/identity-neutral. 3. Missing important comparison with some state-of-the-art many-to-many non-parallel voice conversion work, like AUTOVC: Zero-Shot Voice Style Transfer with Only Autoencoder Loss(ICML 2019). It seems AUTOVC has better reported MOS and higher quality audio samples, although it performs on spectrogram not waveform.

[Author Response · NeurIPS 2019]

# Response to reviews of paper 3681 – "Blow: a single-scale hyperconditioned flow for non-parallel raw-audio voice conversion".

We are very grateful to all reviewers for their positive assessment of our work and their constructive feedback. We have considered all minor remarks and typos and already updated our version of the manuscript accordingly.

Two reviewers comment on the forward-backward conversion procedure and our design hypothesis that the $\mathbf{z}$ space should be condition-free (or identity-neutral in the case of voice conversion). We believe the suggestions made in this regard are very interesting, and thank the reviewers for that. We have acted in two ways to improve our work. On the one hand, we have rephrased the "Forward-backward conversion" section (Sec. 4.3), removing some terms that complicated the understanding of the procedure and improving the wording of the remaining ones. On the other hand, following the reviewers' suggestion, we have performed an additional experiment to bring evidence that the $\mathbf{z}$ space is identity-neutral (the details of this experiment are already included in our current version of the manuscript). Our reasoning for the experiment that, if $\mathbf{z}$ vectors carried information about the speaker, then a powerful-enough classifier on top of $\mathbf{z}$ vectors should be able to perform speaker identification much better than random chance, which is 1.1% for the considered data set and split. In the extreme case, if $\mathbf{z}$ vectors carried as much information about speaker identity as the original $\mathbf{x}$ vectors, the classifier should approach the accuracy of the classical-feature linear classifier we use for the Spoofing metric, which is 99.3% for the considered data set and split. Hence, on top of $\mathbf{z}$ vectors, we trained both a random forest classifier with 50 estimators and a multilayer perceptron with 3 layers, and obtained accuracies of 1.8 and 1.4%, respectively; both marginally above random chance. This gives us an indication that that the $\mathbf{z}$ space has limited speaker identity information or that, in the best of cases, such information is not apparent or complex to extract.

Finally, one reviewer points out that we should compare to "AUTOVC: Zero-shot Voice Style Transfer with Only Autoencoder Loss (ICML 2019)". We thank the reviewer for pointing us out to this very recent and interesting work, and we will include it in our "Related work" section (Sec. 2). At the moment of writing this author feedback response, we are working with the provided code to make it work under our setting in order to, in the future, be able to provide additional results.

[Meta-Review · NeurIPS 2019]

the paper proposes a normalizing-flow based generative model called Blow for non-parallel raw-audio voice conversion that makes use of speaker identity labels. The model makes use of a single-scale structure, conditioning module based on hypernetworks, shared speaker embeddings. Reviewers have recognized the novelty of the paper that consider normalizing flows for voice conversion. They also liked the extensive experiments as well as the ablation studies, and the provided samples of the voice conversions.